# The Role of Long Period Stacking Ordered Phase in Dynamic Recrystallization of a Mg–Gd–Y–Zn–Zr Alloy during Multi-Directional Forging Process

**DOI:** 10.3390/ma13153290

**Published:** 2020-07-24

**Authors:** Huiling Liu, Yingze Meng, Huisheng Yu, Wenlong Xu, Siyang Zhang, Leichen Jia, Guoqin Wu

**Affiliations:** 1Mechanics Institute, Jinzhong University, Jinzhong 030619, China; liuhuiling@jzxy.edu.cn; 2College of Mechanical and Electrical Engineering, North University of China, Taiyuan 030051, China; 3School of Material Science and Engineering, North University of China, Taiyuan 030051, China; yuhuisheng1995@163.com (H.Y.); wenlong@hotmail.com (W.X.); 15536641664@163.com (S.Z.); jlc226688@hotmail.com (L.J.); wuguoqin5656@hotmail.com (G.W.)

**Keywords:** Mg–Gd–Y–Zn–Zr alloy, LPSO phase, multi-directional forging process, dynamic recrystallization, texture

## Abstract

The Mg–Gd–Y–Zn–Zr alloy containing a long period stacking ordered (LPSO) phase was subjected to multi-pass deformation by means of a multi-directional forging process, and the microstructure evolution and the influence of the LPSO phase on its dynamic recrystallization (DRX) were studied. The results showed that multi-directional forging can effectively refine the grain with the DRX fraction increased, and DRXed grains lead to the decrease of the texture intensity, which can significantly improve the mechanical properties of the alloy. The different morphologies of the LPSO phase have different degrees of promotion relative to DRX behavior. The lamellar LPSO phase with kinks promoted dislocation plugging, where there could be a potential nucleation site for DRX grains. The fragmented lamellar LPSO phase promoted the DRX process through the particle-stimulated nucleation mechanism, and the block-shaped phase was more prone to stress concentration, which promoted DRX. These effects resulted in continuous grain refinement and a more uniform microstructure.

## 1. Introduction

Now that the resources and the environment have become the primary issues for sustainable development, lightweight has become an important development trend in aerospace, automotive, equipment, and other fields [1]. Under such a development background, the pursuit of a high-strength and high-stiffness transmission case structure will inevitably increase the transmission assembly mass, and a lightweight design of the transmission case is the general trend [2]. Magnesium alloys have excellent specific strength and specific stiffness, excellent shock absorption performance, high thermal conductivity, electrical conductivity and good shielding, and they have become the first choice as metal structural materials for the lightweight design of various load-bearing components such as various boxes or shells used in automobiles [3]. Most of the box or shell parts currently use casting technology. During the casting process, the existence of casting defects reduces the strength and toughness, which makes it difficult to meet the design requirements [4]. At present, with the development trend of a lightweight overall structure design, the strength and toughness of boxes or shells need to be higher and higher. If high-performance forged blanks are used for subsequent extrusion or processing, the design requirements can be met [5]. In order to obtain higher mechanical properties in the subsequent extrusion process, the material and blank forming process are particularly important. Large pre-strain using severe plastic deformation (SPD) can effectively refine grains and improve the mechanical properties of the alloy [6].

The SPD technology mainly includes equal channel angular pressing (ECAP) [7], circulating closed die forging (CCDF) [8], high-pressure torsion (HPT) [9], and other processes, which can all produce strong grain refinement effect and make the grains reach the nanometer level. However, these processing methods are usually only suitable for the laboratory preparation of samples and are not suitable for large-scale industrial production. Multi-directional forging (MDF) has been confirmed by materials such as Mg-RE [10,11] because of its simple process to be applied to bulk alloys, which can achieve SPD and thus effective grain refinement.

The addition of rare earth elements can also improve the mechanical properties of Mg alloys to obtain high specific stiffness, high temperature strength, and creep resistance [12]. The addition of Zn to the Mg-RE alloys will produce the precipitation of long-period stacking ordered (LPSO) phase, which can effectively improve the microstructure characteristics of the alloy [13]. LPSO usually has two forms: a block-shaped phase and a lamellar phase [14,15]. Its special kinking phenomenon adapted to deformation can effectively enhance the strength and ductility of the alloy [16,17,18,19]. Mg–Gd–Y–Zn–Zr alloy with an LPSO phase is an important type of high-strength heat-resistant magnesium alloy, which has excellent room-temperature and high-temperature properties [20]. It can be applied to parts that work for a long time at 200–300 °C, which have broad application prospects in aerospace, transportation, weapons, and other fields [21]. This alloy when used in the production of transmission cases can meet the normal working requirements of automobiles, and it will achieve a low-carbon and lightweight production of transmission assemblies [22]. However, its poor formability hinders its application range [17]. In order to overcome the problem of poor processing performance, the Mg–Gd–Y–Zn–Zr alloy is usually subjected to severe plastic deformation (SPD) at high temperature. The grain refining caused by dynamic recrystallization can further improve the mechanical properties of the alloy.

Chen et al. [23] studied the microstructure and mechanical properties evolution of an Mg97Y2Zn1 alloy with an LPSO phase after large deformation by ECAP, and they found that after 6 passes, the mechanical properties of the alloy were significantly improved, whose yield strength (YS) and ultimate tensile strength (UTS) reached 400 MPa and 450 MPa respectively, while the block-shaped phase is prone to cracking under large plastic deformation, resulting in a decrease in elongation. Zhang et al. [24,25] studied the effect of ECAP deformation on Mg–Gd–Zn–Zr and Mg–Y–Zn–Zr alloys, showing that ECAP deformation can significantly improve the strength of the alloy, which is mainly caused by the kink of the LPSO phase. Lu et al. [26] obtained the Mg97.1–Gd1.8–Zn1–Zr0.1 alloy with nano-scale grains by ECAP deformation, with a tensile strength of 387 MPa and an elongation of 23%, indicating that a uniform and fine microstructure can increase both strength and elongation. Therefore, it is of great significance to study the change of LPSO phase and its effect on DRX during deformation.

Some scholars have conducted some research on it before. Wu et al. [27] studied the Mg–Zn–Y alloy under compression and pointed out that the kink boundary of the LPSO phase tends to become a nucleation site for dynamic recrystallization due to the high energy stored. Zhou [28] et al. obtained the LPSO phases of different morphologies by homogenizing Mg–6.9Gd–3.2Y–1.5Zn–0.5Zr (wt %) alloy at 783 K for different durations. The effect of dynamic recrystallization in the compressed experiments was studied, and it was pointed out that the internal thin-platelet LPSO has a more obvious promotion effect on dynamic recrystallization (DRX) near the α-Mg grain boundary. Chen et al. [29] analyzed the compression of Mg–Zn–Y alloy under different conditions and pointed out that the LPSO phase has a promoting and pinning effect relative to DRX. However, most of these studies have focused on simple stresses, and there have been few studies on LPSO relative to DRX under the complex stress conditions of process production.

The effect of the MDF process under decreasing temperature on the microstructure and texture of Mg–Gd–Y–Zn–Zr alloy has been studied before [22], while the effect of the LPSO phase on the grain refinement during the MDF process is not yet clear. In order to further reveal the specific influence of the LPSO phase during the MDF process on the dynamic recrystallization behavior, the isothermal MDF process was selected to deform the Mg–9.45Gd–3.28Y–1.77Zn–0.34Zr alloy containing the LPSO phase, investigating the evolution of the microstructure and mechanical properties under different passes. Furthermore, the influence of different morphologies of LPSO on the dynamic recrystallization behavior was further analyzed.

## 2. Materials and Methods 

The experimental alloy is an Mg–9.55Gd–3.28Y–1.77Zn–0.34Zr alloy made by semi-continuous casting. Before deformation, it is homogenized at 520 °C for 20 h and cooled in air. The sample is a cube sample of 150 mm × 150 mm × 150 mm. The forging experiment was carried out by a hydraulic machine with a limit load of 1250 tons and a forging speed of 5 mm/s.

Before the start of the experiment, the mold and the sample were simultaneously heated to 440 °C and kept for 30 min to ensure that the temperature was uniform. The MDF experiment forged the alloy from three different directions at 440 °C, that is, the direction of forging was gradually rotated by 90° (that is, from A to B to C). Each forging was called a pass, and the strain of each pass was 0.5. The experiment process and observation position are shown in Figure 1.

After grinding and polishing, the samples were corroded with an etching solution of 0.5 g picric acid, 2 mL of acetic acid, 2 mL of distilled water, and 14 mL of alcohol, after which they were observed with an optical microscope (OM) of Zeiss Axio Imager A2m. The backscattered electron (BSE) of a Hitachi SU5000 at 20 kV was used to observe the microstructure. The texture evolution was studied by electron backscatter diffraction (EBSD). Before the EBSD experience, the sample was polished using multifunctional ion milling, with an operating voltage of 6.5 kV for 30 min. The directional imaging microscope (OIM) software is used for subsequent processing of the EBSD data, and the grains with grain orientation spread (GOS) < 2 are regarded as DRXed grains. In order to determine the tensile properties, dog-bone tensile specimens were processed and sampled in parallel to the forging direction, and the five specimens were averaged in the same state. The tensile test was carried out at room temperature by an INSTRON 3382 tensile machine with a tensile rate of 1 mm/min.

## 3. Results

### 3.1. Microstructure of the Initial Alloy

Figure 2 shows the backscattered electron (BSE) image of the microstructure of the alloy after homogenization. Lamellar phases are densely arranged inside the grains. These lamellar phases have been confirmed as 14H structures in our previous studies [30]. Adding Y and Zn elements to the magnesium alloy will increase the base stacking fault of the alloy [31]. When the alloy is heat-treated, the solute atoms diffuse into the stacking fault to form a periodic arrangement of 14H [32,33]. The block-shaped phases are distributed at the grain boundaries, where most of the edges are smooth, and some of the block-shaped phase edges have needle-like protrusions toward the inside of the grain (the yellow dotted frame in Figure 2b). Bright precipitates are distributed at the edge of the block-shaped phase and at the grain boundaries (red and black arrows in Figure 2b). By energy dispersive spectrdmeter (EDS) analysis, the Gd + Y atomic ratio reached 86%, which is the (Gd, Y) rich rare earth phase (Figure 2c). This phase is generated during solidification, or when the 14H–LPSO phase transitions and dissolves in the solid solution state [30]. The grains are relatively uniform, and the average grain size measured by the intercept method is 196 µm.

### 3.2. Microstructure Evolution of the MDFed Alloys

Figure 3 shows the OM image of the microstructure under different passes. It can be observed that in 1 pass, the alloy grains are unevenly distributed, and the large lamellae phase still exists. The size of some large grains reaches 200 µm. Compared with the initial state, the block-shaped phase distribution in the grain boundary is obviously broken (dark blue arrow in Figure 3a). Obvious kinks can be observed in the lamellar phases within the grains (shown by the yellow rectangular frame in Figure 3a), and some of the lamellar phases are broken along the kinks (light blue dotted line in Figure 3a). The dynamic recrystallized grains are distributed in a necklace shape along the initial grain boundary (the red dotted frames in Figure 3a), forming a typical bimodal microstructure. 

As the number of deformation passes increases, the grain size of large grains containing lamellar phases drops to about 100 µm at 2 passes, the grain boundaries are mostly jagged, and the edge lamellar phase breakage increases. The remaining lamellar phases all have different degrees of kinks (the yellow rectangular frames in Figure 3b). The dynamic recrystallization grains increased obviously, and the microstructures were more uniform (the red dotted frames in Figure 3b). The block-shaped phase breaks down with the size reducing. The semi-continuous network structure is not maintained, and the distribution is more uniform at the grain boundaries.

At 3 passes, the amount of deformation increases, the grains continue to refine, and the size of the large lamellar phase decreases. The remaining lamellar phase can still be observed to adapt to the deformation, but the kink angle is smaller (the yellow rectangular frames in Figure 3c). The block-shaped phase is refined into small blocks, which are more diffusely distributed (dark blue arrows in Figure 3c). It is conducive to dislocation plugging, which activated dynamic recrystallization in a larger range (the red dotted frames in Figure 3c).

In order to show the change of LPSO phase in the alloy more clearly, we conducted BSE observation on the alloy under different passes (Figure 4). The upper right corner is an enlarged view of the position of the white box in the figure. The matrix is shown in dark gray, the lamellar phase is distributed inside the grains, and the block-shaped phase shows a brighter contrast. After 1 pass, the inner lamellar phase is kinked obviously, the kink angle is the largest, which is up to 59.4°, and the edge is rarely broken. The block-shaped phase has a unique microstructure, which is distributed in a semi-continuous network at the grain boundary, and there are many needle-like protrusions on the edge, which kinked under the action of deformation.

After 2 passes, the size of the block-shaped phase is reduced, and the edge extension is more likely to be broken (black arrow in Figure 4b). It is easy to cause a stress concentration around the block-shaped phase, and the edge of the large lamellar phase is broken, which is evenly distributed in the matrix (black oval dotted frame in Figure 4b).

At 3 passes, the edge of the block-shaped phase is not smooth and breaks. Some of the block-shaped phases are broken into dense small masses due to the stress concentration (as shown by the black arrow in the Figure 4c), similar to the phenomenon we observed in repetitive upsetting-extrusion before [34]. The large lamellar phase is reduced, and the kink phenomenon is weakened, indicating that the strain of the alloy is more uniform under this condition. In a larger range, the small lamellar phases are evenly distributed in the matrix, which further stimulates dynamic recrystallization and promotes grain refinement.

Figure 5 and Figure 6 study the texture evolution and grain size changes of MDFed alloys in different passes. As the number of deformation passes increases, the dynamic recrystallization fraction increases significantly from 0.284 to 0.703, and the low-angle boundary (LAGBs, white lines protruding) score decreases, indicating that the increase in the degree of deformation significantly enhances the DRX process. In Figure 5, the color inside the large grain changes significantly, indicating that the internal lattice rotation adapts to the deformation, and the surrounding small dynamic recrystallization shows a random color orientation.

The intensity of (0001) and (1120) pole figures in Figure 5 decreases with the increase of deformation passes. Studies have shown that DRX can significantly reduce the deformation texture of the alloy, indicating that the DRX degree is enhanced and the deformation texture is weakened under a higher strain [15]. Through grain size analysis, as the number of deformation passes increases, the grain size distribution becomes more concentrated, and the average grain size continues to decline from 106.02 μm in 1 pass to 15.12 μm in 3 passes (Figure 6a). The size of the unDRXed original grains decreased significantly with the increase in the amount of deformation, and the size of the dynamic recrystallization also decreased, indicating that DRX continued to occur and the grains continued to be refined (Figure 6b,c).

### 3.3. Mechanical Performance

Figure 7 shows the mechanical properties of the alloy at different processing passes measured by tensile tests at room temperature. The average value of ultimate tensile strength (UTS), yield strength (YS), and failure elongation (FE) of the samples are shown. In order to more clearly represent the scatter of the YS, UTS, and FE after each forging pass distribution, we apply the ANOVA method to calculate the variance, and the results are shown in Table 1. At one pass, the average UTS, TYS, and FE values of the alloys were 245 Mpa, 192 Mpa, and 7.3%, respectively. As the number of passes increases, the total tensile strength, yield strength, and elongation increase. The average values of UTS, TYS, and FE at 2 passes were 318.3 Mpa, 230.3 Mpa, and 13.67%, respectively. After 3 passes, the mechanical properties of the alloy slightly improved and reached the highest level, and the average value of UTS, TYS, and FE increased to 325.3 MPa, 233 MPa, and 14.67%.

Although the difference in the average mechanical properties of 2 or 3 passes is not obvious, the scatter is different. According to the variance in Table 1, the distribution of the mechanical properties of the alloy is more dispersed after 2 passes, the variance of which is the largest, reaching 127.81, 142.36, and 1.56 respectively—that is, the mechanical property data of the samples in multiple tensile test experiments after 2 passes are not stable and there are large fluctuations. 

This is because although the fraction of dynamic recrystallization continues to increase with the increase in the number of deformation passes, at the second pass, complete recrystallization has not occurred with about a 0.5 recrystallization fraction. From an overall point of view, the grains continue to be refined, but due to certain differences in the amount of deformation at different positions, there are differences in the local grain sizes. The more dynamic recrystallized grains appear in the region with greater strain, the more obvious the grain refinement, which leads to better mechanical properties. With the difference in the sampling position of the tensile test bar and the difference in the grain size in different regions, the performance data distribution is more scattered and the variance is large, which will bring great hidden dangers in the subsequent processing and production. 

The variance of the alloy mechanical properties data at 1 pass is less than that at 2 passes because the dynamic recrystallization just begins to occur. The alloy presents a bimodal structure with a few dynamically recrystallized grains surrounding the coarse grains, and the similarity of the structure at different locations change little. Therefore, although the mechanical performance is relatively low at one pass, the mechanical property data does not fluctuate much.

With the further increase of MDFed passes, the variance of the mechanical properties of the 3-pass alloy has decreased significantly; especially, the variance of YS has decreased significantly from 142.36 in 2 passes to 37.65 in 3 passes. As the dynamic recrystallization fraction increases to 0.7, the grains are refined in a wide range, and the structure become more uniform. The mechanical properties become more stable with the increase of the MDFed passes, indicating that the mechanical properties of the alloy after 3 passes were further improved.

Combined with its microstructure, it can be concluded that as the amount of deformation increases, the dynamic recrystallization fraction increases, the dynamic recrystallization range increases, and the structure is more uniform. The increase in alloy mechanical properties is attributed to the refinement of the grains. The increase in the number of fine grains leads to an increase in the grain boundary, which causes grain boundary strengthening. It is also conformed to the principle of the Hall–Petch equation, improving the performance.

## 4. Discussion

The occurrence of DRX is always associated with multiplication, tangle, annihilation, and a recombination of dislocations [35]. In the observation of dynamic recrystallization behavior, it was found that LPSO phases with different morphologies have different induction effects relative to dynamic recrystallization. The role of the lamellar phase with kinks, the fragmentation of the lamellar phase, and the block-shaped phase in the DRX behavior are discussed separately.

### 4.1. Lamellar LPSO Phase with Kinks Induced Dynamic Recrystallization

Figure 8 further analyzes a deformed grain with kinks in the sample at 1 pass, and the dynamic recrystallization behavior is observed more clearly. In the deformed grain, the band-shaped regions with gradual color change can be observed, which are kinks of the lamellar LPSO phase. Along the black arrows AB and CD, the misorientation between the adjacent points and the point-to-origin line graph (Figure 8c,d) shows that the lattice of the grain continuously rotates due to the deformation. The lattice orientations of different kinked regions are different (hexagonal prisms in Figure 8a), indicating that a large number of dislocations in the kink are active.

Dynamic recrystallized grains are observed at the black dotted line in Figure 8. In this region, the misorientation between adjacent points increases (Figure 8c), and the distribution of LAGBs is concentrated. The kernel average misorientation (KAM) value around the dynamic recrystallized grains is maintained at a high level, and the internal KAM value is reduced (Figure 8b). As the strain increases, the dislocation density increases and moves to form LAGBs. The orientation difference continues to increase and gradually changes to the HAGBs, forming dynamic recrystallized grains, which reduces the local stress concentration.

A large number of LAGBs are found at the kink region, as shown by the red arrows in the Figure 8, which are usually formed by the dislocations accumulated during the deformation process [31,36]. Comparing the KAM diagram (Figure 8b), the twisting at the same position makes the local KAM value increase, which can further indicate that the density of dislocations is higher in this region, causing plugging. Kinks can significantly change the orientation of matrix grains, forming a significant orientation gradient and promoting the formation of DRXed grain boundaries at the kink boundaries. These regions can be inferred as potential nucleation sites for dynamically recrystallized grains. The study of Hess and Barrett [37] showed that the formation of dislocations with opposite signs caused kinks, and the movement of dislocations accumulated at some special positions, gradually forming kinks with an obvious kink interface. At the same time, the accumulation of dislocations is also conducive to the formation of dynamic recrystallization, that is, the kink promotes the formation of dynamic recrystallization by promoting the dislocation plugging. Zhou et al. attributed the development of DRX at the kink boundary of the Mg–Gd–Y–Zn–Zr alloy to the typical continuous DRX (CDRX) mechanism [38].

### 4.2. Fragmentation of Lamellar LPSO Phase Induced Dynamic Recrystallization

The regions with a large lamellar LPSO phase and fragmented lamellar LPSO phase were further investigated to reveal its effect on DRX behavior (Figure 9). The large lamellar LPSO phase area is framed by the blue dashed line, and the fragmented lamellar phase is framed by the yellow dashed line (Figure 9a). It can be seen that a large amount of dynamic recrystallization occurs in the region filled with the fragmented lamellar phase, and the large lamellar phase region still maintains the deformed initial structure (Figure 9b). This stimulates the nucleation mechanism through typical particles. The fragmented lamellar LPSO phase in the matrix hinders the movement of dislocations, making the dislocations easier to entangle, forming a misorientation gradient area, where dynamic recrystallization is easy to occur—that is, the particle-stimulated nucleation (PSN) occurs [39]. This fully proves that the fragmented lamellar phase promotes the occurrence of dynamic recrystallization. The original grains accumulate a higher density of dislocations due to deformation, and the dislocations in the surrounding DRX area form dynamic recrystallized grain boundaries to obtain a lower KAM value. At the same time, it can be observed that the KAM value of the area where the original grain edge and the dynamic recrystallization interface is high (Figure 9c), which indicates from the side that the dislocation can be pinned by the LPSO phase, thereby further promoting the DRX process.

### 4.3. Block-Shaped Phase Induced Dynamic Recrystallization

Figure 10 shows the grain reorientation diagram of dynamic recrystallization in the sample under different processing passes, which is superimposed with the SEM picture, showing the distribution of the block-shaped phase in the alloy, and also more clearly showing the role of the block-shaped phase in the dynamic recrystallization in the deformation. It can be observed from Figure 10 that the block-shaped phases are distributed at the grain boundaries of the dynamic recrystallization region (blue arrow in Figure 10), and they are almost completely surrounded by dynamic recrystallization. Studies have shown that stress concentration is more likely to occur around the block-shaped phase, which is conducive to the plugging of dislocations and becomes the preferential nucleation site for new grains, promoting the occurrence of dynamic recrystallization [34]. At 1 pass (Figure 10a), the phenomenon is more obvious; DRX tends to occur near the block-shaped phase, but only a small amount occurs inside the large lamellar phase area, indicating that the block-shaped phase plays a greater role in promoting the dynamic recrystallization process than the lamellar phase with kinks. As the number of deformation passes increases, the range of dynamic recrystallization gradually increases, and the kinked lamellar phases, fragmented lamellar phases, and block-shaped phases together promote dynamic recrystallization to make the alloy structure more uniform and improve its mechanical properties (Figure 10b,c).

## 5. Conclusions

In this paper, the MDF experiment of the homogenized Mg –Gd–Y–Zn–Zr alloy at 440 °C was carried out to systematically study the microstructure and properties of the alloys of different passes, and it focuses on the effect of LPSO phases with different morphologies on dynamic recrystallization behavior. The conclusions are as follows:(1)As the number of processing passes increases, the dynamic crystallization fraction increases, and the grains are refined. After 3 passes, the dynamic recrystallization fraction reaches 0.703. The grain size decreases significantly from 106.02 to 15.13 µm, and the structure is more uniform.(2)Dynamic recrystallization weakens the texture and significantly improves the mechanical properties. In 3 passes, the texture intensity is reduced to 2.62, the tensile strength is increased to 325.3 MPa, and the elongation increased to 14.67%.(3)The kinked lamellar phase rotates the grains, promotes dislocation plugging, and forms grain boundaries more easily, which is a potential nucleation site for dynamically recrystallized grains; the fragmented lamellar phase promotes the dynamic recrystallization process through the particle-stimulated nucleation mechanism, and stress concentration occurs around the block-shaped phase to promote dynamic recrystallization, making the microstructure more uniform.

## Figures and Tables

**Figure 1 materials-13-03290-f001:**
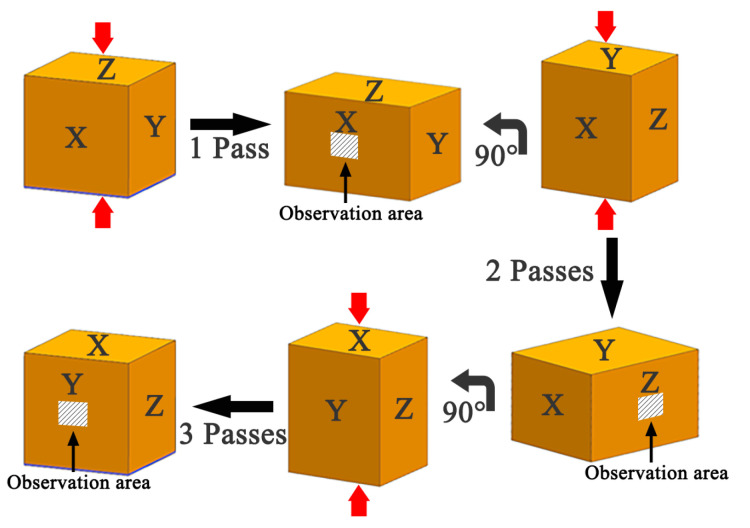
Experimental process and observation location.

**Figure 2 materials-13-03290-f002:**
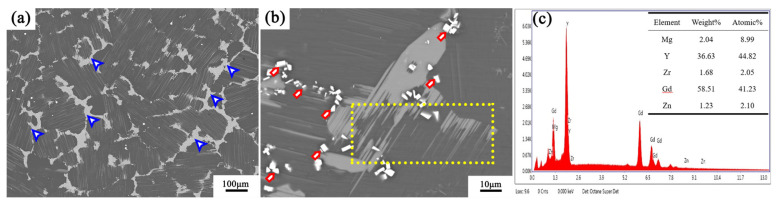
Backscattered electron (BSE) images of the alloy after homogenization: (**a**) overall microstructure; (**b**) block-shaped phase; (**c**) EDS result of the precipitate

**Figure 3 materials-13-03290-f003:**
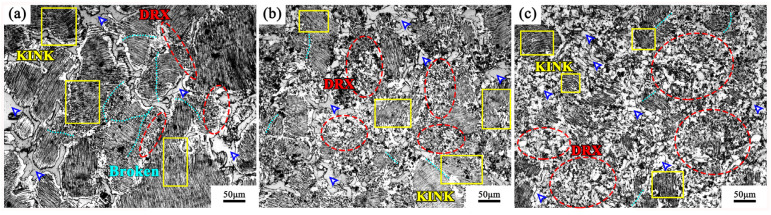
Optical microscope (OM) images of the alloys under different multi-directional forging (MDF)ed passes: (**a**) 1 pass; (**b**) 2 passes; (**c**) 3 passes.

**Figure 4 materials-13-03290-f004:**
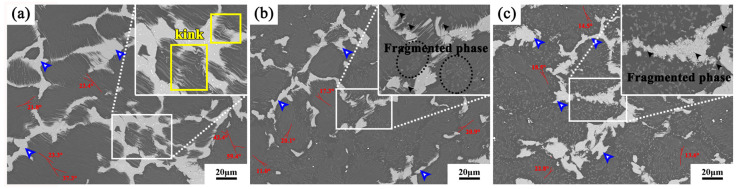
Backscattered electron (BSE) images of the alloys under different MDFed passes: (**a**) 1 pass; (**b**) 2 passes; (**c**) 3 passes.

**Figure 5 materials-13-03290-f005:**
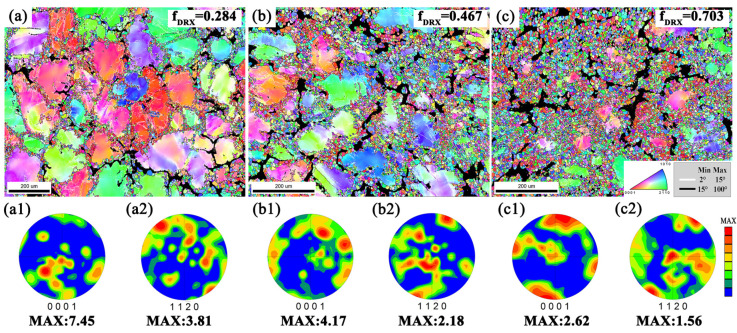
Directional imaging microscope (OIM) images and corresponding (0001) and (1120) pole figures of the alloys under different MDFed passes: (**a**) 1 pass; (**b**) 2 passes; (**c**) 3 passes.

**Figure 6 materials-13-03290-f006:**
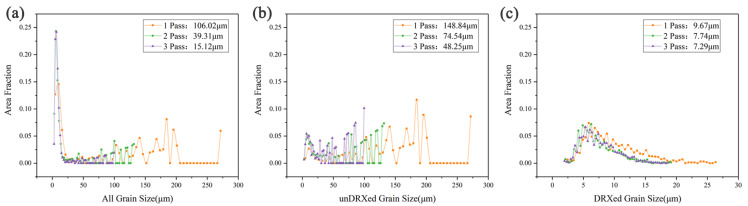
Grain size of the alloys under different MDFed passes: (**a**) all grains; (**b**) unDRXed grains; (**c**) DRXed grains.

**Figure 7 materials-13-03290-f007:**
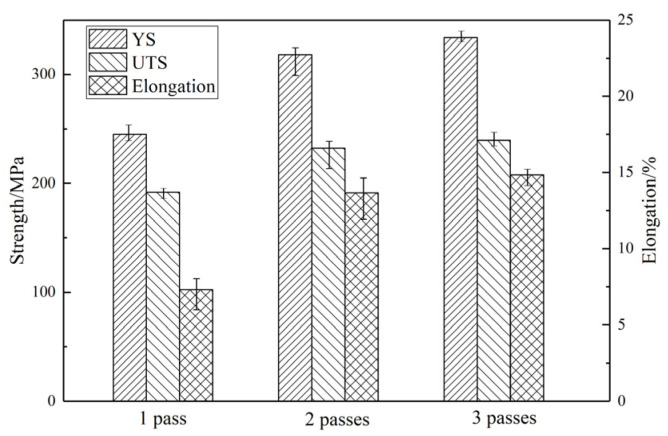
Mechanical properties of the alloys under different MDFed passes.

**Figure 8 materials-13-03290-f008:**
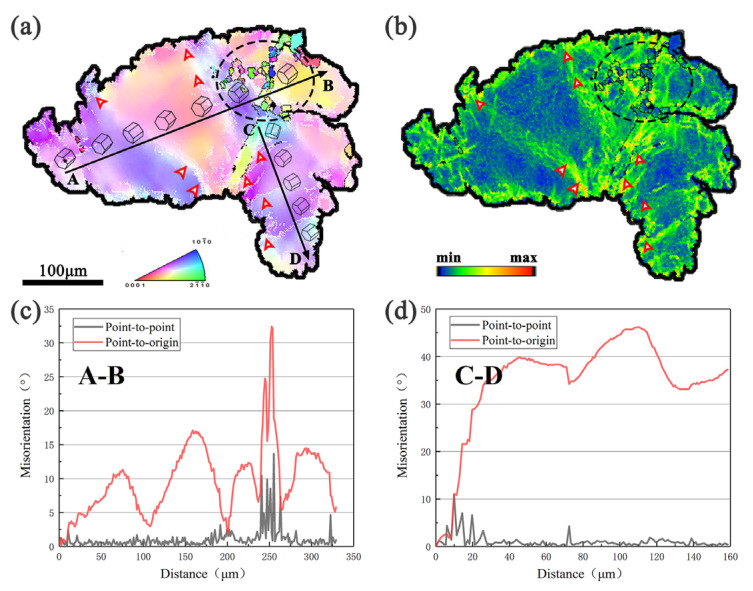
The typical kinked grain selected in the sample at 1 pass: (**a**) OIM map; (**b**) kernel average misorientation (KAM) distribution map; (**c**) line graph of misorientation angle along the AB in (**a**); (**d**) line graph of misorientation angle along the CD in (**a**).

**Figure 9 materials-13-03290-f009:**
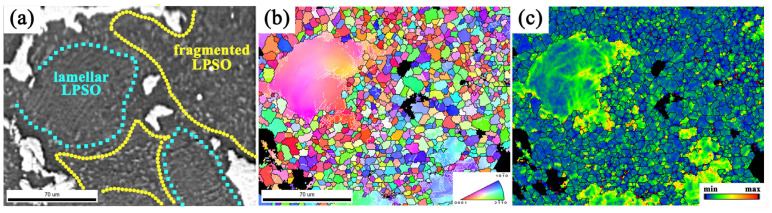
The typical regions with a large lamellar phase and fragmented lamellar phase: (**a**) SEM image; (**b**) OIM map; (**c**) KAM distribution map.

**Figure 10 materials-13-03290-f010:**
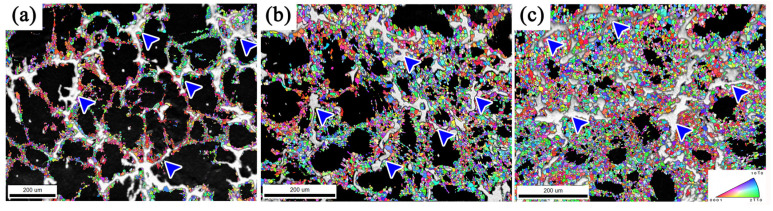
The regions with block-shaped phase and DRXed grains: (**a**) 1 pass; (**b**) 2 passes; (**c**) 3 passes.

**Table 1 materials-13-03290-t001:** The scatter of the yield strength (YS), ultimate tensile strength (UTS) and failure elongation (FE) under different MDFed passes.

Variance	UTS	YS	FE
1 pass	92.89	49.62	1.29
2 passes	127.81	142.36	1.56
3 passes	64.22	37.65	0.22

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
