# Peer review of "The Role of Long Period Stacking Ordered Phase in Dynamic Recrystallization of a Mg–Gd–Y–Zn–Zr Alloy during Multi-Directional Forging Process"

_materials, 2020, doi:10.3390/ma13153290_

Round 1

Reviewer 1 Report

Thanks a lot for a very interesting and accurate work!

Some remarks are listed below:

  1. What is the average grain size after each pass? In the Section 3.2, the size of the large grains are mentioned while in the Figure 6(a), the average grain size is given? Moreover, in the conclusion other numbers are listed (from 106.02 mkm to 15.13 mkm). Please specify.
  2. The mechanical properties after second and third passes are quite similar. How it could be explained when the different grain size and different amount of DRXed grains are taken into account?

Author Response

Thanks for your comments. The specific reply to you is in the attachment, please check.

Reviewer 2 Report

The article is rich in microstructural results, but very poor in results linked to the material performance. The authors discuss in the first paragraph of the intruductory section the application field of Mg-alloys, but it will take a long way to come from the results of this article to the actual application of the studied alloy in the automotive industry. Despite the microstructure of the alloy after multi-directional forging is well described and its evolution through the different forging directions is presented I really miss the applied value of these results (my background is mechanical engineering linked to the automotive industry). To augment the applied value of the article for the engineering applications the results in section 3.3 (Mechanical properties) should be presented in a more complete manner:

1.) What is the scatter of the YS, UTS and elongation after each forging pass? Are the differences after 2nd and 3rd pass staistically significant?

2.) How do sig-eps diagrams look for the virgin material and after each forging pass?

3.) What is the pocedure to transform the block of of Mg-Gd-Y-Zn-Zr alloy into the gearbox body? Milling? Die casting? At which manufacturing phase should the MDF be applied?

4.) When talking about the powertrain applications (gearbox body), what is the resistance of this alloy against the (vibrational) fatigue?

I would expect that the authors make the opinion about the above-mentioned issues. Othervise this article is just a good exercise in using an expensive experimental equipment.

Author Response

(The authors gave the same response as above.)

Reviewer 3 Report

Dear Authors, 

Please find my comments attached in the pdf along with this message. 

Thank you. 

Author Response

(The authors gave the same response as above.)

Round 2

Reviewer 2 Report

Dear authors,

To a large extent I am dissatisfied with the improvements in the article due to the following reasons:

1.) I specifically asked for a sig-eps diagrams of tensile tests, but were not added.

2.) I specifically asked for statistical significance of the differences in material properties. They are not estimated subjectively, but the statistical tool ANOVA should be used instead.

3.) I asked you for the opinion on how do you imagine to apply this process for manufacturing of the complex product os the gearbox body is. You didn't add comments on that topic, which makes the first paragraph of the introduction completely unnecessary. It actually misleads the readed.

If this issues are not improved my next decision Will be reject.

Author Response

Thank you very much for your suggestions. Our reply to you has been uploaded to the attachment, please check .
